# Civil society perspectives on tuberculosis care for people living with HIV in Brazil: A study informed by Social Representations Theory

Gabriela Tavares Magnabosco[1,2]☯*, Fernanda de Paulo Pedroso[1]☯,
Isadora Gabriella Silva Palmieri[1]☯, Letícia Baio de Souza[1]☯, Heitor Hortensi Sesnik[1]☯,
Sidnei Nathan Soares Turquino[1]☯, Ketlyn Andriele Lomes da Cruz[2]‡,
Renato Meggiato Nabas[2]‡, Heloísa do Carmo Antonio[2]‡,
Márcio Vinícius Ferreira Resende[2]‡, Gabriel Pavinati[2]☯

1 Postgraduate Program of Nursing, State University of Maringá, Maringá, Paraná, Brazil, 2 Department of Nursing, State University of Maringá, Paraná, Brazil

☯ These authors contributed equally to this work.
‡These authors also contributed equally to this work.
* gtmagnabosco@uem.br

## Abstract

In Brazil, tuberculosis–human immunodeficiency virus (TB–HIV) coinfection remains a major public health challenge despite advances in antiretroviral therapy and tuberculosis preventive treatment (TPT) Civil society has historically contributed to HIV responses, but little is known about how it perceives TB care for people living with HIV (PLHIV). This study examined how organized civil society perceives and represents TB care for PLHIV. We conducted a qualitative study guided by Social Representations Theory, with five focus groups involving 37 representatives from civil society organizations in five Brazilian state capitals in 2025. Three thematic categories were identified: (1) social and institutional neglect of TB, evidenced by the absence of campaigns, delayed diagnosis, and shortages of medications in some localities; (2) stigma, poverty, and social exclusion in the context of coinfection, in which TB–HIV coinfection was described as a factor that exacerbates these phenomena; and (3) civil society and non-governmental organizations (NGOs) as mediators of care, which act as a bridge between socially vulnerable populations and health services. Participants acknowledged that persistent barriers to TB care are further intensified by the presence of HIV. This study advances current knowledge by explicitly framing civil society not only as a mediator, but as a co-producer of TB–HIV care, particularly in contexts of social vulnerability.

## Introduction

Tuberculosis (TB) remains a major global public health challenge, ranking among the leading causes of morbidity and mortality worldwide and having regained, in 2023,

**Data availability statement:** Due to the qualitative nature of the study and the sensitivity of the data, transcripts cannot be publicly shared. Data are available upon reasonable request and subject to ethical approval. Requests for access should be directed to the Research Ethics Committee of the State University of Maringá (email: copep@uem.br). The committee will evaluate requests and ensure compliance with ethical standards. Data are securely stored at the State University of Maringá and will be maintained for long-term availability in accordance with institutional policies.

**Funding:** This work was supported by the National Council for Scientific and Technological Development (CNPq), Brazil (grant number 445760/2023-0). GTM is the principal investigator and coordinator of the multicenter research project that generated this manuscript. GP received a technological development fellowship as a doctoral researcher affiliated with the project. FPP received a master's fellowship and contributed to study coordination, focus group facilitation, data collection, transcription, and qualitative data analysis. RMN and KALC received undergraduate research (scientific initiation) fellowships and participated in focus group facilitation and transcription of audio-recorded data. The funders had no role in study design, data collection and analysis, decision to publish, or preparation of the manuscript.

**Competing interests:** The authors have declared that no competing interests exist.

its position as the leading cause of death from a single infectious agent [1,2]. When associated with Human Immunodeficiency Virus (HIV), TB-HIV coinfection assumes an even more complex profile, characterized by elevated illness and mortality [3]. People living with HIV (PLHIV) are estimated to be 23 times more likely to develop active TB compared with the general population [3].

Brazil is included on the World Health Organization (WHO) list of countries with a high burden of TB–HIV. Nationally, the proportion of TB-HIV coinfection among new TB cases rose from 9.5% in 2023 to 11.4% in 2024, the highest level recorded in the historical series [3]. Progress has been documented through the expansion of HIV testing and the increase in antiretroviral therapy (ART) coverage, which reached 57.3% among PLHIV diagnosed with TB in 2024. Tuberculosis Preventive Treatment (TPT) also advanced: in 2024, 56,079 cases were registered, including 20.9% of PLHIV [3].

Despite these improvements, Brazil is still distant from meeting the goals established by the End TB Strategy and the National Plan to End TB [3]. Some challenges remain to eliminate TB-HIV coinfection: diagnostic delays, shortages of supplies, low TPT coverage, and interruptions in care, compounded by structural determinants such as poverty, food insecurity, institutional racism, and state neglect [4,5]. In addition, intersecting stigma associated with coinfection reinforces exclusion, social isolation, and treatment abandonment [6,7].

Organized civil society has played a central role in shaping health and social responses [8], encompassing collectives, forums, networks, and non-governmental organizations (NGOs) engaged in rights advocacy, social oversight, and community mobilization. In Brazil, this role has been particularly prominent in the consolidation of social programs and public policies, with historical milestones dating back to the establishment of the Unified Health System (SUS), the world's largest universal public health system [8].

From this premise, the trajectory positions civil society as a key analytical lens for understanding the TB–HIV response, as these actors operate at the interface between affected communities, health services, and public policies, especially in contexts marked by inequality and stigma. Making it possible to identify institutional gaps, coping strategies, symbolic and material barriers, and opportunities for the co-production of care. Accordingly, this study aimed to understand how organized civil society perceives and represents TB care for PLHIV.

Civil society perspectives were adopted as an analytical benchmark because these actors operate at the interface between communities, health systems, and policy implementation, allowing a situated assessment of how TB–HIV responses function in practice.

## Methods

### Study design

This was an interpretive qualitative study using focus group discussions (FGDs), theoretically informed by the Social Representations Theory (SRT). The theory

conceptualizes social knowledge as socially constructed and shared, guiding practices and communication. We operationalized SRT by examining processes of anchoring and objectification, that is, how TB and TB–HIV coinfection was interpreted using pre-existing social categories (anchoring) and transformed into concrete meanings and practices in participants' narratives (objectification) [9,10]. For example, anchoring occurred when participants interpreted TB through familiar social categories such as poverty or marginalization, while objectification was evident when these meanings materialized into practices such as avoidance or delayed care seeking. Consolidated Criteria for Reporting Qualitative Research (COREQ) was followed (see S1 Table) [11].

## Setting and participants

Data were collected between January and July of 2025 in five Brazilian capitals representing the country's macro-regions: Manaus (North), Recife (Northeast), Campo Grande (Central-West), Rio de Janeiro (Southeast), and Porto Alegre (South). Cities were selected intentionally considering TB and TB-HIV coinfection burden, historical relevance of social mobilization, and sociocultural and geographic diversity.

Participant selection was purposive, aiming to include representatives of organized civil society engaged in HIV or TB public policies, including forums, networks, NGOs, and community collectives. The representants spoke from both personal and professional experiences, reflecting on their own lived experiences as well as interactions with affected people, sharing perceptions about care practices and institutional challenges.

Eligibility criteria were: (i) age ≥ 18 years; (ii) residence in the capital or its metropolitan region; (iii) fluency in Portuguese; and (iv) active involvement in HIV or TB advocacy or social oversight. Exclusion criteria included absence from scheduled sessions or disengagement from representative roles at the time of data collection. Recruitment was carried out by trained members of the comunity facilitators using contact lists and community referrals. Participant characteristics and epidemiological indicators of each study site are presented in S2 Table.

## Data collection

Data were collected through in-person focus groups. FGDs were conducted in Portuguese at accessible institutional venues, with only moderators and participants present. Group size ranged from six to twelve participants, lasting approximately 60 minutes. Sessions followed a semi-structured guide with open-ended questions; the main guiding question was: "What are your perceptions of TB diagnosis and treatment in PLHIV?" Complementary questions explored experiences, challenges, and strategies of care. The full discussion guide is provided in S1 Text.

FGDs were moderated by civil society representatives with prior engagement in HIV/TB advocacy. Facilitators received a two-hour standardized training from the research team, covering study objectives, non-directive moderation, equitable participation, and bias mitigation. Potential positionality and power dynamics were addressed by clarifying objectives, ensuring confidentiality, and promoting balanced discussion. Continuous support was available from the academic research team throughout data collection.

Moderators also kept brief field notes. Anonymity was ensured by coding participants with the prefix "P" followed by a number and city name (e.g., P1, Recife). Discussions followed a natural flow, without correction or validation of statements, respecting spontaneity. Data collection was concluded when thematic saturation was reached, monitored through a saturation matrix, showing that no new codes emerged after the fifth FGD (see S3 Table) [9,12].

## Data analysis

All FGDs were audio-recorded and transcribed verbatim. Transcripts were read in full and analyzed using thematic content analysis as described by Minayo [12], integrated with SRT. Coding combined inductive categories emerging from the data and deductive categories based on SRT [10,11,13]. Two researchers independently coded the transcripts and met regularly to reconcile differences, ensuring analyst triangulation.

Themes were then interpreted through SRT, identifying anchoring and objectification processes, and mapping central versus peripheral elements of the representations [10,13]. Memos and an audit trail documented analytical decisions. The final codebook and thematic categories are presented in S4 Table.

## Ethics statement

The study complied with Brazilian National Health Council Resolution 466/12 and was approved by the Research Ethics Committee of the State University of Maringá (Opinion No. 7,348,262; CAAE No. 84341924.9.0000.0104). All participants provided written informed consent and were informed about the voluntary nature of participation, anonymity, confidentiality, and the right to withdraw at any time.

## Results

A total of 37 representatives participated across sites with varying TB–HIV epidemiological profiles (see S1 Table). They were distributed across five focus groups conducted in Porto Alegre (n = 6), Rio de Janeiro (n = 6), Campo Grande (n = 9), Recife (n = 9), and Manaus (n = 7). Participants included both women and men, with a predominance of female representation across sites. The age of participants ranged from 22 to 64 years, reflecting the inclusion of individuals at different stages of life and with diverse trajectories of engagement in civil society and community-based responses to TB–HIV. Three thematic categories were identified based on participants' perceptions and experiences: (1) social and institutional neglect of tuberculosis, (2) stigma, poverty, and social exclusion in the context of coinfection, and (3) civil society and NGOs as mediators of care (see S3 Table for full codebook and subcategories).

### Category 1: Social and institutional neglect of tuberculosis

Across focus groups, participants commonly described perceptions and experiences that reflected what they understood as social and institutional neglect of TB, particularly when associated with HIV. Across focus groups, TB was perceived as an "erased" disease, both in public discourse and within health services, contributing to delayed diagnosis, fragmented care, and weakened prevention efforts.

A recurring perception was the lack of knowledge about the persistence and relevance of TB, which participants associated with a sense of abandonment by the State. In several groups, TB was described as a disease that many believed no longer existed, reflecting the absence of sustained public communication. This perception was illustrated by statements such as: *"People think TB no longer exists"* (P3, Porto Alegre) and *"There is a lack of information for people [living with HIV] to become aware of their situation and the risks they are facing"* (P2, Recife).

Participants consistently emphasized the absence of public campaigns, educational actions, and clear care protocols in both specialized services and primary healthcare. This absence was described as a setback compared with earlier periods, reinforcing perceptions of institutional disengagement. As one participant noted, *"There is a lack of television campaigns. We used to see campaigns"* (P6, Rio de Janeiro).

Beyond symbolic invisibility, participants reported concrete forms of neglect in the daily functioning of health services. Participants frequently perceived low clinical suspicion of TB among health professionals, resulting in diagnostic delays and missed opportunities for early treatment. This perception was illustrated by the following account: *"When a person manages to see a doctor, the last hypothesis considered is TB"* (P2, Campo Grande).

Difficulties in accessing diagnostic tests, specialized consultations, and medications were also described as common. Participants reported long waiting times and interruptions in care that compromised timely treatment initiation, as exemplified by the statement: *"The medication is not available immediately. [...] They told him to come back in three months to see if he could get an appointment in pulmonology"* (P2, Recife).

This institutional neglect was also perceived to extend to preventive strategies, particularly TPT. Participants reported that health professionals often failed to offer preventive treatment to PLHIV or did not adequately explain its purpose

and benefits. This issue was illustrated by the following statement: "T*he treatment exists, but it is not offered […] We are invited to receive guidance about what preventive treatment is, and if we want to start it, we do so. However, the person who begins HIV treatment is not presented with this option" (P3, Manaus).*

Although TPT was included in the discussion guide, participants did not spontaneously elaborate on preventive treatment as a central strategy for TB prevention among PLHIV. References to TPT were sparse and primarily limited to reports that it was not routinely offered, poorly explained, or difficult to access within health services. From a SRT perspective, this limited salience suggests that TPT has not been fully anchored in participants' shared representations of TB–HIV care, remaining weakly objectified in everyday practices and narratives.

Within this representational framework, TB was anchored as a "forgotten" or "erased" disease, socially reclassified as a problem of the past. This anchoring helps explain the normalization of diagnostic delays, weakened prevention efforts, and institutional disengagement reported by participants.

**Category 2: Stigma, poverty, and social exclusion in the context of coinfection**

Narratives across groups indicated that TB was symbolically associated with stigma, poverty, and marginalization, especially when occurring alongside HIV. Across focus groups, TB–HIV coinfection was perceived as intensifying social vulnerability and shaping experiences of fear, discrimination, and exclusion from care.

TB was commonly framed as a "disease of poor people," associated with precarious housing, inadequate sanitation, food insecurity, and limited access to social protection. This was illustrated by statements such as: *"If I have TB, people will speak badly of me because I had TB"* (P1, Porto Alegre) and *"TB occurs precisely among those experiencing social vulnerability"* (P4, Recife).

When associated with HIV, TB diagnosis acquired heightened symbolic meaning. Participants described TB–HIV coinfection as amplifying fear, judgment, and emotional distress, often perceived as a sign of clinical deterioration or imminent death. This perception was expressed in accounts such as: *"TB is a breath toward death for those with HIV"* (P5, Rio de Janeiro).

Stigma was described as both external and internalized, contributing to social isolation and treatment discontinuity. Participants reported that fear of discrimination discouraged individuals from seeking care or adhering to treatment. As one participant noted, *"Prejudice and discrimination are among the reasons people abandon treatment"* (P3, Porto Alegre) and *"[...] the professional doesn't care at all [...] they don't even want to provide care or treatment for a person living with HIV—let alone someone with TB"* (P6, Manaus). They also expressed fear regarding preventive treatment, as can be observed in the participant's statement: *"People are generally embarrassed to say that they are going to undergo preventive treatment"* (P2, Porto Alegre).

Socioeconomic precariousness was described as a major barrier to accessing and sustaining care. Participants emphasized that hunger, unstable housing, and the need to prioritize daily survival limited individuals' ability to seek diagnosis and complete treatment. This was reflected in statements such as: *"It's useless to give medication to someone who doesn't have the slightest chance of survival"* (P2, Campo Grande).

Participants also reported experiences of distancing and discrimination within health services. Accounts included professionals avoiding physical contact, rushing consultations, or failing to provide adequate explanations. These practices were perceived as particularly harmful in contexts of extreme vulnerability, as described: *"Do you think a professional, given their rushed schedule, will take the time to talk and explain this to the person [about TB treatment]? No, they won't"* (P2, Porto Alegre) and *"A patient said they would pick up the medication, and the nurse handed it through the window because she didn't want to have contact with a patient with TB"* (P4, Manaus).

Taken together, the findings suggest that TB–HIV coinfection functions as a powerful social marker through which structural inequalities are reproduced within both community dynamics and health-care institutions, underscoring that stigma and material deprivation are not ancillary to care but are embedded in its organization, delivery, and outcomes.

**Category 3: Civil society and NGOs as mediators of care**

Participants reported civil society organizations and NGOs as central mediators of care for people affected by HIV and TB, particularly in contexts marked by institutional gaps and social vulnerability. These organizations were perceived as bridging populations that remained "invisible" to formal health services.

Across FDGs, participants emphasized the role of NGOs in actively reaching individuals who did not regularly attend health units, especially in marginalized territories. As one participant explained: *"They simply don't go [health professionals]. So, as an institution or non-governmental organization operating in the territory, we have this role"* (P1, Rio de Janeiro).

Beyond outreach, participants highlighted the relational dimension of civil society work. NGOs were described as spaces of mutual care, solidarity, and collective learning, where PLHIV supported one another emotionally and practically. These bonds were perceived as essential for sustaining engagement with care, particularly among individuals facing stigma and social exclusion.

Participants also reported that NGOs were often more effective than formal services in engaging people who use drugs and other marginalized groups. Civil society was described as operating through trust-building approaches and partnerships that health professionals were often reluctant or unprepared to adopt: *"We care a lot for each other, we observe each other closely, [...] we greatly support one another"* (P6, Recife) and *"Civil society acts precisely through this partnership"* (P1, Campo Grande).

In addition, NGOs were described as key sources of information about TB, communicating in accessible and culturally appropriate language grounded in lived experience. Participants contrasted these approaches with technical or biomedical discourse, emphasizing their role in facilitating understanding and engagement, as pointed out: *"You build your own language without using the medical, technical, or entirely formal language"* (P1, Rio de Janeiro).

At the same time, participants reported significant challenges faced by NGOs, including limited funding, lack of institutional recognition, and exclusion from formal decision-making spaces. Several accounts described difficulties in establishing dialogue with healthcare managers and being recognized as legitimate partners by health professionals, as commented: *"We have to push for everything; if we don't, we are excluded from the annual education and health programs"* (P6, Manaus); *"[…] the system as a whole does not give us this openness"* (P2, Porto Alegre) and *"Healthcare professionals do not recognize civil society as partners"* (P1, Recife).

Despite these barriers, participants emphasized the political and emotional commitment of community organizations. NGOs were described as safe and welcoming spaces where individuals could be heard, supported, and empowered to claim their rights. Participants also highlighted the historical role of organized civil society in shaping public policies in Brazil, particularly in the HIV response, emphasizing that partnerships between government and community were essential for ensuring equitable and responsive TB–HIV care.

## Discussion

Guided by SRT, this study highlights how meanings attributed to TB-HIV coinfection are collectively constructed. Participants described shared representations in which tuberculosis-particularly when associated with HIV-was framed as a neglected, stigmatizes, and socially determined condition. These representations were not merely symbolic, but materialized in concrete practices, influencing access to services, continuity of care, and engagement with prevention and treatment strategies [14].

Anchoring was evident when TB was associated with "poverty," situating the disease within preexisting social categories, while objectification occurred as these ideas materialized into concrete practices, such as avoidance of contact with people diagnosed with TB or non-adherence to treatment. Understanding these symbolic constructions is essential to reorient care practices and public policies, ensuring that interventions align with meanings shared by affected groups [9,10].

Participants described TB-HIV coinfection as a double burden, intensifying psychological suffering and social isolation. Stigma was not limited to interpersonal interactions but was reinforced by institutional practices, including lack of information, inadequate communication strategies, and discriminatory attitudes from health professionals. These experiences point to the need for attentive listening and recognition of subjectivities in care, echoing studies that emphasize person-centered approaches to strengthen adherence and improve outcomes [15,16].

The narratives also highlighted structural barriers within health services. Reports of delays in examinations, shortages of medications, and fragmented pathways of care indicate persistent weaknesses in TB control, particularly in peripheral territories where resources are scarce [17]. These findings align with existing literature documenting how shortages of human and material resources compromise TB and HIV programs in low- and middle-income countries [18].

Amid these structural barriers, TPT emerged as a fragile and inconsistently incorporated element of TB–HIV care. Although participants recognized TPT as an important preventive strategy—aligned with recommendations from the WHO and Brazil's Ministry of Health—its presence in care trajectories was described as limited, uneven, and largely dependent on individual initiative rather than systematic offering within services [19]. From a SRT perspective, this weak incorporation suggests that TPT remains poorly anchored in shared representations of TB–HIV care, resulting in limited objectification in routine practices. Participants pointed to territorial inequalities in access and significant gaps in information and communication about preventive treatment, reinforcing its marginal role in everyday care experiences.

This fragile incorporation of TPT cannot be understood in isolation from broader social and structural conditions that shape access and continuity of care. In this sense, participants emphasized social protection as a fundamental component for strengthening TB control and sustaining preventive treatment over time. These accounts resonate with studies showing that social protection policies—such as cash transfers, transportation support, and food assistance—are decisive in enabling access and adherence, mainly in contexts of poverty and food insecurity [1,2,20]. Integrating biomedical interventions with social protection is therefore crucial to achieve global elimination targets. It reveals the importance of different sectors, like social assistance and public security.

This perspective aligns with the Sustainable Development Goal 3.3, which aims to eliminate TB and HIV by 2030, and with the National Plan for the End TB, which incorporates social protection as a strategic element to reduce inequalities and improve clinical outcomes [1,2,21]. In this sense, territory was described as a key factor shaping care trajectories. In contexts marked by urban exclusion, food insecurity, and precarious living conditions amplify institutional invisibility and restrict access to timely care [6,22,23]. The isolated action of the health sector has proven insufficient given the complexity of the determinants involved [24,25].

Within this context, civil society organizations emerge as co-producers of TB–HIV care, actively shaping access to services, treatment adherence, and community-based support strategies [1,2,19,20]. Their actions fill institutional gaps and facilitate access for marginalized populations, such as people deprived of liberty, people who use drugs, homeless and residents of favelas [26,27]. The most recent report by the WHO Civil Society Task Force on TB [1] emphasizes that their meaningful participation is crucial for accelerating the achievement of global targets and reducing health inequalities.

It must be considered the need to move beyond ad hoc or informal engagement of community organizations toward their formal integration into TB–HIV programmes. This integration may include contractual or co-management arrangements for community-based prevention activities, treatment adherence support, and stigma reduction initiatives, as well as the systematic involvement of civil society representatives in planning, monitoring, and evaluation processes.

Strengthening institutional mechanisms for participation—such as dedicated funding lines, inclusion in governance structures, and formal recognition of community-based monitoring—can enhance accountability and responsiveness of TB–HIV services. Aligning national TB and HIV strategies with these participatory approaches is aligned with the End TB Strategy and SDG 3.3, and may contribute to more equitable, people-centred, and sustainable responses, particularly for populations experiencing social exclusion.

However, the limited institutional integration of these organizations diminishes their transformative potential. The lack of formal recognition, insufficient continuous funding, and the undervaluation of their contributions hinder their inclusion in decision-making processes. This underscores the need to promote participatory environments that incorporate social oversight and territorial representations in the formulation of health policies [1,2,20,21].

An effective response to TB-HIV coinfection requires the convergence of biomedical interventions, social protection measures, and community-based strategies. Expanding social benefits such as cash transfers and transportation support, combined with rights-based approaches and equity-centered care, represents a feasible and necessary strategy for reducing treatment abandonment and mortality.

In this context, recognizing civil society not only as an ally but as a co-producer of care is indispensable to transform TB-HIV coinfection responses into inclusive, equitable, and sustainable public health policies. Only by institutionalizing the voices and practices of civil society will it be possible to bridge biomedical strategies with social justice, and thereby accelerate progress toward TB and HIV elimination.

## Limitations

Although it included different Brazilian capitals, the country's diversity in sociodemographic and programmatic contexts may shape perceptions differently. The purposive sampling focused on organized civil society representatives, and thus findings cannot be generalized to the broader population. Additionally, participant recruitment and group facilitation were mediated by local representatives, which may have influenced accounts, despite the standardized training and centralized analysis adopted to mitigate potential biases. As with all qualitative studies, the findings are context-specific and intended to generate interpretive insights rather than statistical generalization.

## Conclusion

The SRT analyzed in this study indicate that TB-HIV coinfection continues to be shaped by meanings that reinforce historical vulnerabilities, such as stigma, social exclusion, and programmatic shortcomings. Beyond describing barriers, this study makes a novel contribution by presenting the perspective of organized civil society, revealing how these actors construct and share meanings about healthcare, and how their practices emerge as a concrete response to institutional gaps.

The novelty of this research lies in recognizing community organizations as co-producers of care, not merely as intermediaries between the community and health services, but as key actors in coordinating response strategies that combine support, social protection, active outreach, and the defense of rights. In this regard, the results underscore the urgency of public policies that incorporate the knowledge, practices, and representations of civil society.

Addressing TB-HIV coinfection requires an approach centered on equity, human rights, and meaningful social participation, capable of integrating intersectoral policies and community-based practices to promote inclusive, responsive, and empowering care. By incorporating the perspective of civil society, this study provides valuable insights for the development of health policies that are more attuned to the realities of vulnerable territories and contributes to repositioning social participation as a strategic pillar for TB and HIV elimination.

## Supporting information

**S1 Table. COREQ (COnsolidated criteria for REporting Qualitative research) Checklist.**
(DOCX)

**S2 Table. Epidemiological indicators of study sites and participant characteristics by city.**
(DOCX)

**S1 Text. Focus group semi-structured interview guide.**
(DOCX)

**S3 Table. Saturation matrix of thematic codes across focus group discussions.**
(DOCX)

**S4 Table. Codebook of thematic categories and subcategories.**
(DOCX)

## Acknowledgments

The authors acknowledge the support of the *General Coordination of Tuberculosis, Endemic Mycoses and Non-Tuberculous Mycobacteria*, within the *Department of HIV, AIDS, Tuberculosis, Viral Hepatitis and Sexually Transmitted Infections*, *Health and Environmental Surveillance Secretariat*, *Ministry of Health of Brazil (CGTM/Dathi/SVSA/MS)*, for their institutional support and technical collaboration in the development of this study.

## Author contributions

**Conceptualization:** Fernanda de Paulo Pedroso, Isadora Gabriella Silva Palmieri, Letícia Baio de Souza, Heitor Hortensi Sesnik, Sidnei Nathan Soares Turquino.

**Supervision:** Gabriela Tavares Magnabosco, Isadora Gabriella Silva Palmieri, Gabriel Pavinati.

**Visualization:** Gabriela Tavares Magnabosco, Fernanda de Paulo Pedroso, Isadora Gabriella Silva Palmieri, Letícia Baio de Souza, Heitor Hortensi Sesnik, Sidnei Nathan Soares Turquino, Gabriel Pavinati.

**Writing – original draft:** Gabriela Tavares Magnabosco, Fernanda de Paulo Pedroso, Isadora Gabriella Silva Palmieri, Letícia Baio de Souza, Heitor Hortensi Sesnik, Sidnei Nathan Soares Turquino, Gabriel Pavinati.

**Writing – review & editing:** Gabriela Tavares Magnabosco, Fernanda de Paulo Pedroso, Isadora Gabriella Silva Palmieri, Letícia Baio de Souza, Heitor Hortensi Sesnik, Sidnei Nathan Soares Turquino, Ketlyn Andriele Lomes da Cruz, Renato Meggiato Nabas, Heloísa do Carmo Antonio, Márcio Vinícius Ferreira Resende, Gabriel Pavinati.

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
