## [Decision Letter · Decision Letter 0]

18 Dec 2025

PGPH-D-25-03019

Civil society perspectives on tuberculosis care for people living with HIV in Brazil: a study informed by Social Representations Theory

Dear Dr. Magnabosco,

Thank you for submitting your manuscript to PLOS Global Public Health. After careful consideration, we feel that it has merit but would benefit from some revision. Therefore, we invite you to submit a revised version of the manuscript that addresses the points raised during the review process.

We look forward to receiving your revised manuscript.

Kind regards,

Graeme Hoddinott, Ph.D

Academic Editor

Journal Requirements:

Additional Editor Comments (if provided):

The reviewers have both noted the strength of your submission. They have also both identified areas for improvement that seem reasonable to me. Please address these to the best of your ability or include thorough explanation for why that is not possible / desirable.

Reviewers' comments:

Reviewer's Responses to Questions

**Comments to the Author**

1. Does this manuscript meet PLOS Global Public Health’s publication criteria?

Reviewer #1: Yes

Reviewer #2: Yes

2. Has the statistical analysis been performed appropriately and rigorously?

Reviewer #1: N/A

Reviewer #2: N/A

3. Have the authors made all data underlying the findings in their manuscript fully available (please refer to the Data Availability Statement at the start of the manuscript PDF file)?

Reviewer #1: Yes

Reviewer #2: Yes

4. Is the manuscript presented in an intelligible fashion and written in standard English?

Reviewer #1: Yes

Reviewer #2: Yes

Reviewer #1: Major Strengths

Strong methodological rigor.

The use of focus groups, analyst triangulation, an audit trail, adherence to COREQ, and a saturation matrix demonstrate a high standard of qualitative practice.

Clear alignment with SRT.

The study operationalizes anchoring and objectification effectively and integrates TSR across analysis and interpretation.

Rich, well-supported findings.

The thematic categories are well developed and consistently supported with participant quotes.

High policy relevance.

The manuscript addresses stigma, territorial inequities, social exclusion, and the essential role of civil society in TB–HIV responses—issues of global significance.

Ethical compliance.

Ethical approval, consent, and justified data access restrictions are clearly documented.

Areas for Improvement

1. Abstract is overly dense and should be tightened.

While the content is strong, the abstract reads more like a compressed summary of results and discussion. Shortening sentences and more sharply highlighting the novelty—particularly civil society as “co-producers of care”—would increase impact.

2. Some repetition in Discussion and Conclusion.

Several ideas appear multiple times (e.g., stigma, institutional neglect, equity, civil society’s role). Streamlining these sections would strengthen clarity and flow.

3. Strengthen the articulation of novelty.

The manuscript identifies civil society as a mediator of care and co-producer of response strategies, but this contribution could be stated more explicitly and concisely in both the Discussion and Conclusion.

4. Minor clarity edits.

A few long sentences could be broken up, and some terminology is used interchangeably (e.g., “neglect,” “invisibility,” “erasure”). Standardizing terminology would improve precision.

5. Abstract and Results should avoid language that implies generalizability.

Given the qualitative design, it is important to maintain interpretive accuracy and avoid phrasing that implies statistical generalization.

Minor Editorial Notes

A careful read-through to reduce verbosity in certain paragraphs—especially in Discussion—would enhance readability.

No major grammatical errors were identified, but minor tightening will improve clarity for an international readership.

Overall Recommendation

Minor Revision.

The manuscript is technically sound, the conclusions are well supported by the data, and the study offers meaningful insights into an understudied dimension of TB–HIV care. The required revisions relate primarily to clarity, conciseness, and sharpening the manuscript’s unique contributions. Once these issues are addressed, the paper will be very well suited for publication in PLOS Global Public Health.

Reviewer #2: Review of Title: Civil society perspectives on tuberculosis care for people living with HIV in Brazil: a study informed by Social Representations Theory

Summary of research and overall impression

This paper introduces an important topic by examining civil society perspectives on TB and HIV, particularly in a context such as Brazil, where civil society has played a significant role in shaping responses to both epidemics. The study is generally well written and draws on an appropriate methodological framework. However, revisions are needed to strengthen the introduction and results sections in particular, with more minor revisions required in the methods and discussion sections.

Abstract

The abstract is clear and well-written, providing a detailed overview of the study.

Introduction

The introduction would benefit from revisions to more clearly set up the topic, including a review of its overall structure to improve logic and flow. For example, the authors could begin with a brief global overview of TB and TB/HIV co-infection, including advances in ART and TPT alongside persistent challenges, before moving to a more focused discussion of the role of civil society in HIV responses. This could then be contextualised within Brazil, outlining the HIV/TB situation nationally, the specific role of civil society actors, and why civil society provides an important lens—or “thermometer”—for understanding health and social responses in this setting. The introduction should conclude with a clear statement of the study aims.

In addition, the authors should carefully review specific phrasings for accuracy. For instance, the opening sentence states that TB has been a leading cause of morbidity and mortality since 2023; however, TB has been a leading cause of morbidity and mortality for many years prior to 2023, and this should be corrected.

Methods

The methods section is well written, with appropriate attention to relevant methodological detail. The overall approach is methodical and thoughtfully designed. However, a few minor revisions are required to improve clarity and structure.

I suggest revising the section headings to better introduce the different components of the methods: Study design and setting; Sampling and recruitment; Data collection; Data analysis; and Ethics statement (with the ethics statement moved to the end of the methods section).

Under Study design and setting, the authors include a definition of TSR; however, additional detail would be helpful on what is meant by “examining processes of anchoring and objectification,” as this concept is central to the data analysis and interpretation and is referred to later in the manuscript. If available, a figure illustrating this framework could strengthen this section. This subsection should also include a description of the study setting, and the heading should be renamed accordingly.

Please include specific months in the text (e.g., “between January and July 2025”) rather than using date formats such as 29/01/2025. In addition, the description of site selection could be rephrased as “cities were purposively selected based on/considering …”.

Under a separate heading, Sampling and recruitment, in addition to the information currently provided under Participants, the authors should describe how civil society organisations were selected. If selection was purposive, this should be explicitly stated, along with details on how participants within these organisations were recruited or invited to participate. Provide details to outline who they are speaking on behalf of, as in the results, they seem to be speaking from their own experiences (of care), as well as recounting specific patient stories, and so on.

The Data collection section could be moved earlier in the methods (e.g., at line 100), with minor revisions for consistency. This is otherwise a clear and well-written section.

Finally, the Data analysis section should explicitly state that interviews were transcribed verbatim, as this information is currently included under Data collection but is more appropriately located under Data analysis.

Results

General revisions

For each results category, consider adding a brief introductory statement under the heading to orient the reader and explain how the category was derived from participants’ accounts, before moving into illustrative detail. For example, under Category 1, the authors could begin with a framing sentence such as: “Participants commonly described perceptions that reflected the social and political erasure of TB …”

Throughout the results, statements should be revised to ensure that findings are clearly presented as participants’ perceptions and experiences rather than as factual claims (e.g., “participants reported…,” “participants described…,” “participants felt…”). Where relevant, please indicate whether a sentiment was commonly expressed, raised by several participants, or mentioned by only one or a few participants.

Greater contextualisation is needed to integrate the excerpts from the FGDs. At present, there is an overreliance on direct quotations, with insufficient analytic framing. Consider introducing excerpts more explicitly (e.g., “as illustrated in the following FGD excerpt…”), and, where appropriate, supplementing with interview excerpts to strengthen interpretation.

Please review verb tense throughout the results section to ensure consistent use of the past tense rather than the current present tense, as these findings reflect perceptions and experiences reported at a specific point in time (in the past).

Finally, all FGD excerpts should be reviewed for relevance and length, with unnecessary words removed where possible. As noted above, the main text should foreground general patterns and analytic interpretations, with quotations used selectively to add nuance or illustrate key points rather than to carry the analysis.

Specific revisions

The first two sentences of the results section could be merged, as the second sentence does not currently stand alone.

The reference to Supplementary File S3 should be revised to S4 (line 145).

In the first paragraph of Category 1, please revise or expand the explanation of the reported lack of knowledge and sense of abandonment by the state, as this theme is currently underdeveloped and not sufficiently substantiated.

ART and TPT are introduced earlier in the manuscript and included in the topic guide, yet they feature only marginally in the results as modes of HIV and TB prevention. Please clarify whether participants discussed how ART as prevention of HIV and TPT as prevention of TB have influenced perceptions of these illnesses, or explicitly note if these themes did not emerge in the data.

The term territory should be clearly explained for an international readership, as its meaning within the Brazilian context may not be widely understood.

Under Category 3, please clarify whether participants identified any threats to civil society organisations or NGOs, particularly in relation to funding, sustainability, or organisational longevity.

Discussion

The discussion is generally well written and thorough. However, the links between the study findings and broader patterns of perceptions and experiences need to be articulated more clearly earlier in the manuscript (particularly in the methods and results) and then explicitly revisited in the opening paragraph of the discussion.

As noted in the comments on the results, the discussion expands on TPT and treatment as prevention, yet these themes are only marginally reflected in the results section. Please ensure closer alignment between the results and discussion, either by strengthening the presentation of these findings in the results or by tempering their emphasis in the discussion.

Conclusion

There is substantial repetition of similar points across the discussion and conclusion sections. The conclusion would benefit from being streamlined to one to three concise paragraphs that clearly summarise the key messages and contributions of the study, rather than reiterating points already covered in detail in the discussion.

**Do you want your identity to be public for this peer review?** For information about this choice, including consent withdrawal, please see our Privacy Policy

Reviewer #1: **Yes:** Agus Fitriangga

Reviewer #2: **Yes:** Hanlie Myburgh

---

## [Decision Letter · Decision Letter 1]

8 Feb 2026

PGPH-D-25-03019R1

Civil society perspectives on tuberculosis care for people living with HIV in Brazil: a study informed by Social Representations Theory

Dear Dr. Magnabosco,

Thank you for submitting your manuscript to PLOS Global Public Health. After careful consideration, we feel that it has merit but does not fully meet PLOS Global Public Health’s publication criteria as it currently stands. Therefore, we invite you to submit a revised version of the manuscript that addresses the points raised during the review process.

EDITOR:

The reviewers have raised reasonable suggestions for further editing to strengthen the manuscript.

We look forward to receiving your revised manuscript.

Kind regards,

Graeme Hoddinott, Ph.D

Academic Editor

Journal Requirements:

Additional Editor Comments (if provided):

Reviewers' comments:

Reviewer's Responses to Questions

**Comments to the Author**

Reviewer #1: All comments have been addressed

Reviewer #2: (No Response)

publication criteria?

Reviewer #1: Yes

Reviewer #2: Yes

3. Has the statistical analysis been performed appropriately and rigorously?

Reviewer #1: N/A

Reviewer #2: N/A

4. Have the authors made all data underlying the findings in their manuscript fully available (please refer to the Data Availability Statement at the start of the manuscript PDF file)?

Reviewer #1: Yes

Reviewer #2: No

5. Is the manuscript presented in an intelligible fashion and written in standard English?

Reviewer #1: Yes

Reviewer #2: Yes

Reviewer #1: This manuscript provides a quality qualitative research project on the civil society view of tuberculosis care among people living with HIV in Brazil. The topic is prescient and highly applicable to international public health, specifically in the areas of equity, stigma, social participation and the role of community-based actors in TB–HIV responses. We feel that the study fits well within the scope and values of PLOS Global Public Health, with thoughtful acknowledgment throughout” of key works in other languages. –Ed.] Your work makes a substantial contribution by foregrounding civil society as an active actor in care rather than a passive mediator.”

The methods are sound and well described. The application of focus groups in several regions of Brazil enhances the transferability of results, and the use of Social Representations Theory is explicit and strongly embedded throughout analysis. Ethical aspects are well covered and reporting conforms to COREQ guidelines. Reflexivity, saturation check, and analyst triangulation contribute to the study rigor and trustworthiness.

The Finding section is detailed and well proven with material quotes although some aspects of the themes are repeated between the categories, causing little redundancy. Consolidating overlap descriptions and minimizing repetitive quotes would help with the flow of the narrative while still allowing for analytic depth.

Background The description of the theory in the Methods section-- and especially anchoring and objectification—is theoretically sound but a bit lengthy. Succincting this part might, of course, make it more readable and at the same time would not threaten to take away from the theory that underlies our study.

D The Discussion is closely related to the results and contextualizes the findings within available literature and global policies, including the End TB Strategy and SDG 3.3. To enhance the applied relevance of the manuscript, the policy implications might be more explicit. A clear discussion of how civil society organisations could be more formally incorporated into TB–HIV programmes (e.g., through prevention, adherence support, stigma reduction or monitoring) would increase the utility of these findings.

Lastly, a minor language edit is needed mainly to reduce wordiness and repetition of key words. This is generally a well-written paper that could benefit from some clarification, brevity and the translation of policy.

Reviewer #2: Thank you to the authors for the revised manuscript. The paper has improved, and with some further revisions it could be ready for publication. I outline the remaining issues below, focusing mainly on clarity and positioning.

Introduction

Given that the paper centres on civil society, the introduction needs to more clearly introduce and foreground their role in the HIV response. At present, it is still not fully clear how or why civil society is being used as a key lens or benchmark for assessing the response. For an international readership that may be unfamiliar with the Brazilian context (and with civil society’s role in the HIV response more broadly), it would be helpful to include:

• a clear definition of what is meant by “civil society” in this paper,

• an overview of how civil society has been involved in Brazil’s response, and

• a brief explanation of why these perspectives are important for understanding and interpreting the response overall.

Relatedly, the research question and overall aim of the paper would benefit from further clarification, including what is meant by the phrase “to validate the perspective of civil society.”

Methods

Study design:

• In line 72, please clarify whether “It” refers to SRT or to the study itself.

• Lines 77–81 repeat the content of lines 75–77. Rather than restating what anchoring and objectification are, it would be more useful to provide a concrete example to make these concepts more accessible to readers. For instance, an example of “social meaning” and how people might “objectify” this within their everyday social world would help, as these concepts are unlikely to be familiar to many readers.

Results

• Please include brief participant details in the results, as currently presented in S1 Table (incorrectly referred to in the text as S2 – line 144). Consider moving this table into the main manuscript rather than the supplement. At minimum, it would be useful to report the overall age range, male/female number/proportion, and if available, years involved in HIV/TB services or civil society (experience).

• The inclusion of TPT in the results is very welcome and clearly illustrates its limited rollout and familiarity.

• The concluding paragraph at the end of Category 1, which highlights TB as being anchored as a forgotten or erased disease, is strong and important. If possible, consider adding similar concluding reflections for Categories 2 and 3.

Discussion

The discussion and conclusion have been revised well overall. A few minor points remain:

• Line 294 introduces TB as being associated with “dirt,” which does not appear in the results section. If the authors wish to retain this point, it may be helpful to reference existing literature that has documented this association.

• Lines 287–293 make largely the same point across three sentences and could be tightened.

• Lines 324–329: the paragraph on social protection does not clearly follow from the preceding discussion of TPT. A stronger linking or transition sentence would help the flow here.

Other points

• Please check acronyms throughout (e.g., SRT, TB, PLHIV), as their use is currently inconsistent.

**Do you want your identity to be public for this peer review?** For information about this choice, including consent withdrawal, please see our Privacy Policy

Reviewer #1: No

Reviewer #2: **Yes:** Hanlie Myburgh

---

## [Decision Letter · Decision Letter 2]

27 Feb 2026

Civil society perspectives on tuberculosis care for people living with HIV in Brazil: a study informed by Social Representations Theory

PGPH-D-25-03019R2

Dear Dr Magnabosco,

We are pleased to inform you that your manuscript 'Civil society perspectives on tuberculosis care for people living with HIV in Brazil: a study informed by Social Representations Theory' has been provisionally accepted for publication in PLOS Global Public Health.

Best regards,

Graeme Hoddinott, Ph.D

Academic Editor

Reviewer Comments (if any, and for reference):

Reviewer's Responses to Questions

**Comments to the Author**

Reviewer #1: All comments have been addressed

Reviewer #2: All comments have been addressed

publication criteria?

Reviewer #1: Yes

Reviewer #2: Yes

3. Has the statistical analysis been performed appropriately and rigorously?

Reviewer #1: N/A

Reviewer #2: N/A

4. Have the authors made all data underlying the findings in their manuscript fully available (please refer to the Data Availability Statement at the start of the manuscript PDF file)?

Reviewer #1: No

Reviewer #2: No

5. Is the manuscript presented in an intelligible fashion and written in standard English?

Reviewer #1: Yes

Reviewer #2: Yes

Reviewer #1: We appreciate the opportunity to see the revised version of this manuscript.

The authors have addressed the points raised in previous rounds very carefully and comprehensively. There is a notable improvement to the structure, conceptual clarity, and analytical coherence of the manuscript. The Introduction now has a clearer flow from the global TB–HIV context to the Brazilian setting and the unique role of organized civil society. The aim of the study is clearly stated and fits well with the theoretical model.

The Methods section is transparently reported and sufficiently detailed for a qualitative study. The description of Social Representations Theory, such as anchoring and objectification is now clearer and better integrated into the analytic strategy. Description of sampling, recruitment and dual positionality of participants is adequate. Thematic content analysis, independent coding, analyst triangulation and documentation of analytic decisions lend methodological rigor.

The Results section is effectively partitioned into three thematic categories, with exemplary excerpts that build the interpretive claims. Redundancy has been trimmed relative to earlier versions, and the categories are described in short interpretative framing at the beginning and at their end, which enhances readability. The Discussion place the findings in the context of broader TB–HIV, social protection and territorial equity. The framing of civil society as co-producers of care is clear and adds to the literature.

The Manuscript Language is clear, literate and written in academic prose. I did not see any serious grammatical or stylistic issues to be a barrier to comprehension.

The authors describe ethical restrictions concerning access to qualitative focus group data due to the sensitive nature of this type of information, and also specify conditions for accessing the qualitative data. And, though it does not amount to unqualified public access, the rationale seems rooted in considerations of participant confidentiality common to qualitative research.

In conclusion, the manuscript presents methodological rigor, theoretical coherence and policy relevance. In my opinion it qualifies for publication in PLOS Global Public Health.

Reviewer #2: Well done to the authors on well written and researched paper.

**Do you want your identity to be public for this peer review?** For information about this choice, including consent withdrawal, please see our Privacy Policy

Reviewer #1: No

Reviewer #2: No
